# Predictive Capacity Planning for Mobile Networks—ML Supported Prediction of Network Performance and User Experience Evolution

**Igor Tomic** [1,2,*]**, Eoin Bleakley** [3] **and Predrag Ivanis** [1]

1 School of Electrical Engineering, University of Belgrade, 11070 Belgrade, Serbia; predrag.ivanis@etf.bg.ac.rs
2 Aspire Technology Unlimited, 11000 Belgrade, Serbia
3 UCD School of Electrical and Electronic Engineering, University College Dublin, D04 V1W8 Dublin, Ireland; eoin.bleakley@gmail.com
* Correspondence: igor.tomic@aspiretechnology.com

**Abstract:** Network performance prediction is crucial for enabling agile capacity planning in mobile networks. One of the key problems is predicting evolution of spectral efficiency in growing network load conditions. The main factor driving network performance and spectral efficiency is reportedly the Channel Quality Indicator (CQI). In this paper, the performance of different Machine Learning (ML) models were examined, and XGBoost was selected as the best performing model. Furthermore, to improve modeling accuracy, several features were introduced (operating frequency band, Physical Resource Block (PRB) utilization in surrounding cells, number of surrounding cells within a radius, heavy data factor and higher order modulation usage). The impact of these features on CQI prediction were examined.

**Keywords:** mobile network; 5G-NR; LTE; user experience; performance modeling; prediction; CQI; Multiple Input Multiple Output (MIMO); network load; XGBoost

## 1. Introduction

Capacity planning for mobile networks has been a challenge for network planners over the past decade. Traffic in mobile networks has grown exponentially. The growth rate has varied by market, but on average traffic has doubled every two years [1]. This pace of development mirrors Moore's Law. In parallel with growing load, network performance dynamically changes, with expected performance downgrade [2] in cases where there is a lack of investments in additional capacity. Simultaneously, user demand, or more precisely application demand, for throughput and latency has only increased. Further complexity is added by the constraint that the process of adding capacity to mobile networks comes with long cycles. Mobile operators typically need six months to add a 4G or a 5G layer, and two years to build a new base station. Finally, there is always huge pressure to justify all Capital Expenditure (CapEx) investments. In such circumstances, predictive planning is a must. The decision-making process on capacity addition needs to be based on the accurate estimation of future network performance, and what-if evaluations of different scenarios of traffic growth, network performance and capacity expansions.

The problem of predicting user experience in terms of data throughput in mobile networks of the fourth and fifth generation (4G and 5G), based on Orthogonal Frequency Division Multiple Access (OFDMA) techniques, can be decoupled into two parallel streams. Spectrum assets of typical mobile operators are spread over channels in different frequency bands. Channel bandwidth in Long Term Evolution (LTE) systems is 5, 10, 15 or 20 MHz, while in 5G it can be 50–100 MHz in lower frequency bands, and up to 400 MHz on higher frequency bands. Both LTE and 5G systems have resource grids deployed over channels, where the available spectrum is split into Resource Blocks (RBs). In LTE each RB has a

size of 180 kHz, in 5G the size is flexible with a value between 180 kHz and 1440 kHz, depending on use case/numerology [3].

User data throughput in such systems is driven by two components. The first is the number of available resource blocks for each user, which depends on network density (number of base stations deployed in area of interest), user density (number of users to be served in area of interest) and deployed capacity (number of frequency channels and their bandwidths used by base stations on both 4G and 5G technology), resulting in the number of available resource blocks to be shared between users. The second important driver is spectral efficiency of the system, measured as achievable throughput per RB.

The focus of the research presented in this paper is modeling spectral efficiency, and its evolution over time with dynamic network load changes. It is important to understand that patterns of changes are specific for each mobile operator, as they depend on various factors. Some of the most relevant are spectrum assets, network grid-topology and density, quality of radio design and implemented radio solutions, network maturity, user distribution and traffic mix. For that reason, instead of searching for one model that fits all networks, the goal is rather to build frameworks and methodologies that could be applied to different operators, considering their specifics, and capturing network signatures. Mobile communication systems provide a very advanced performance measurement capability. Counters cover different events in the network and various metrics are available. Machine Learning techniques are used to analyze relevant indicators, learn from the past and predict the future.

## 2. Related Work

Previous implementation of Machine Learning in this field [4,5] was mainly related to problems of system improvement (beamforming, multi-user detection, coding/decoding schemes, RF characterization . . . ), network anomaly detection and traffic prediction, or resource management, spectrum allocation and handover prediction. When it comes to capacity planning problems, and the implementation of deep learning techniques, the majority of published studies deal with problems of traffic forecasting. Some of the most recent very relevant examples are [6–10], where various techniques were tested with a better or worse accuracy achieved, depending on the ability to capture different phenomena, like seasonal changes, abnormal events, or changes in the network configuration. However, in the capacity planning process, traffic forecasting can be further mapped only to the first component of user experience discussed in the introduction chapter, which is the number of available resource blocks per user. The second important factor of user experience, which is spectral efficiency, to the authors' knowledge, was not addressed so far. At the same time, in a situation when pretty much all mobile operators are dealing with a growing traffic load and data tsunami, the evolution of spectral efficiency and its accurate prediction seem very relevant problems to address. Novelty introduced with this paper is Machine Learning application, with the aim to capture network signatures and improve spectral efficiency estimation accuracy in increasing network load conditions.

## 3. Theoretical Background

Mobile communication systems, also called mobile cellular systems, as networks are built with large numbers of base stations, where each base station covers some geographical zone. The zone is typically split into three cells. The fundamentals of telecommunications tell us that radio channel performance and capacity are driven by Signal to Noise and Interference Ratio (SINR). The signal level is driven by base station antenna radiated power, and signal propagation loss on the way from the base station to the mobile phone/modem. Noise is the thermal noise, driven by channel bandwidth. Interference is the received unwanted signal in the channel. Interference can be internal—from the same system but dedicated to other users—and external—from other systems. The downlink performance of mobile systems of the fourth and fifth generation—LTE and 5G NR—is mainly driven by internal interference. Both systems are based on OFDMA, where users in the same

cell do not use the same part of the frequency band, unless there are multi-user Multiple Input Multiple Output (MIMO) techniques implemented. MIMO's impact on performance will be very limited, even in the coming years. Hence, the main source of interference is a received signal dedicated to other users, where the dominant component is inter-cell interference suffered from the neighboring cells.

The quality of the radio channel in mobile systems is very different across cells, from close to the antenna to the cell edge, and dynamically changes in time, so LTE and 5G NR schedulers need to adjust parameters of connection to obtain the best possible user experience. This process is called Link Adaptation. In Link Adaption on a Time to Transmit Interval (TTI) basis with 1 ms granularity, the amount of data scheduled during TTI, and modulation and the MIMO scheme are selected to maximize the amount of data to be transmitted and the probability of the successful decoding of the data.

The process is based on reports that the User Equipment (UE) sends to the base station (Figure 1), with information about the quality of a channel, defined by a 3GPP standard as a Channel State Information report (CSI). It has three major components: a Channel Quality Indicator (CQI), Precoding Matrix Index (PMI), and Rank Indicator (RI).

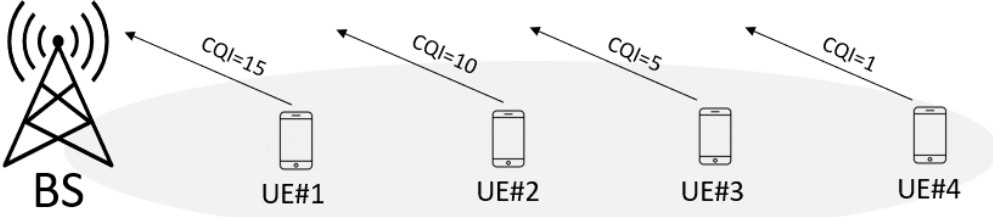

**Figure 1.** CQI reporting in a cell.

CQI is the most important from a link adaptation perspective, as it implicitly indicates a suitable downlink transmission data rate, i.e., the highest Modulation and Coding Scheme (MCS) value at which the UE will be able to properly demodulate and decode the downlink data at the target Block Error Rate (BLER) of 10%. CQI is measured in discrete values between 0 and 15, where index 15 indicates the best channel quality and index 0 indicates the poorest channel quality [10], while the mapping of CQI to the Modulation for the 256QAM capable terminals, defined by 3GPP, is presented in Table 1 (where the coder rates presented in the table are multiplied with factor 1024).

**Table 1.** Link adaptation and mapping of CQI to modulation for 256QAM capable terminals.

| CQI Index | Modulation | Code Rate ×1024 | Efficiency |
|:---------:|:----------:|:---------------:|:----------:|
| 0 | | Out of order | |
| 1 | QPSK | 78 | 0.1523 |
| 2 | QPSK | 193 | 0.3770 |
| 3 | QPSK | 449 | 0.8770 |
| 4 | 16QAM | 378 | 1.4766 |
| 5 | 16QAM | 490 | 1.9141 |
| 6 | 16QAM | 616 | 2.4063 |
| 7 | 64QAM | 466 | 2.7305 |
| 8 | 64QAM | 567 | 3.3223 |
| 9 | 64QAM | 666 | 3.9023 |
| 10 | 64QAM | 772 | 4.5234 |
| 11 | 64QAM | 873 | 5.1152 |
| 12 | 256QAM | 711 | 5.5547 |
| 13 | 256QAM | 797 | 6.2266 |
| 14 | 256QAM | 885 | 6.9141 |
| 15 | 256QA | 948 | 7.4063 |

The Rank Indicator is another important factor for spectral efficiency estimation, as depending on the MIMO performance, there will be roughly parallel streams in spatial multiplexing scenarios. Some of the first performance analysis results from the field are pointing to a better spectral efficiency in 5G systems, compared to older 4G, which is mainly achieved with a higher Rank Indicator usage and better MIMO performance in terms of spatial multiplexing. The most probable reason behind it is the growing number of antenna elements typical for 5G radio system.

## 4. Problem Definition and Research Methodology

The problem of a spectral efficiency prediction is, therefore, the problem of the good estimation of the CQI and Rank Indicator. In previous work [11], authors examined the correlation between CQI and MIMO utilization, while the focus of this paper is CQI modeling. Most of the research in this field is focused on network simulators and performance models that are generic, with some assumptions in the model that can limit the model's accuracy. Furthermore, they are typically focused on system and algorithm design, and rarely deals with network planning problems [12–15]. The reason for this is academics and researchers are rarely involved in industry closely enough to have access to data from real networks. Very few papers and research studies addressing CQI, that combine theoretical and empirical approaches are available. At the same time, in the network evolution process from Global System for Mobile communications (GSM) and 3G towards LTE and, lately, 5G-NR systems, network grid locations used for base stations, as well as some main characteristics of network design, such as antenna height and azimuth, do not change much. For this reason, a lot of network performance modeling can be done based on historical data and recognition of traffic patterns, and network signatures. The authors' approach to the problem of network performance prediction is based on collecting data from commercial mobile networks, operating with huge data sets, and building algorithms and models that can be applied to any network, while capturing its own peculiarities and signatures.

Data collection in a mobile network is done though Performance Management—PM systems. The standard of 3GPP defines how counters are collected. While each Radio Access Network (RAN) vendor has some level of freedom to build a data collection process around different events in the network, differences between RAN solutions in design of PM systems are not significant, and the main indicators are similar. The main indicator used in this study is naturally CQI reporting, analyzed as Cumulative Distribution Function (CDF) during different periods—parts of the day/week/month, cell by cell. The evolution of the CQI reporting process with changes in network load, measured as RB utilization in serving and surrounding cells, are analyzed.

### 4.1. Selection of Features for Improved Algorithm Accuracy

The goal of the study was to build a model that predicts CQI in growing network load circumstances. The main idea was to segment cells according to important factors and drivers for performance. The features examined that could improve modeling accuracy are:

- frequency band,
- Physical Resource Block (PRB) utilization in surrounding cells,
- number of surrounding cells within a given radius,
- heavy data factor,
- higher order modulation usage.

A natural segmentation is by the frequency band, since radio propagation is different at different frequencies, and as discussed earlier, the inter cell interference is the main performance factor. Hence, performance in cells operating at lower frequencies will be more affected by interference in growing network load scenarios. This is due to better radio propagation and a higher probability that the signal from a neighboring cell will interfere with the signal from the serving cell.

Another relevant indicator is the number of neighboring cells within a certain radius. This indicator reflects the network density. The idea, examined previously in [16,17], is

that the denser a network the more sensitive to interference it will be. The average PRB utilization (determined for all surrounding cells and averaged) and total PRB utilization (aggregated for all surrounding cells, encapsulating the number of neighbors in a total score) were both investigated. Surrounding cells were defined as cells within a chosen radius. A radius of 2.5 km, 5 km, and 10 km were compared. Performance was best using a 5 km radius definition of surrounding cells, although the performance using 2.5 km was similar. A 10 km radius led to a substantially worse performance than 5 km or 2.5 km.

Finally, the load in neighboring cells, measured as average PRB utilization, is defined to segment parts of the network with higher traffic. Higher order modulation usage was selected as an indicator that should capture the distribution of users in space and segment cells by the average radio condition. It is defined as the percentage usage of 64QAM and 256QAM. For cells with users distributed closer to the antenna, and in good radio conditions it will have higher values. The last indicator considered is the heavy data factor, with the idea of segmenting cells by the type of traffic. Some use cases, such as video streaming services or heavy FTP download, utilize the LTE/5G-NR network resources to a much larger extent than other use cases, with a small amount of bursty data such as reactions on social media, or messaging applications. Capturing traffic burstiness is not an easy task with available counters in PM systems. The authors have tried to capture this with the ratio of data payload in the cell and PRB utilization. However, a high level of correlation between both the heavy data factor and higher order modulation features with spectral efficiency may be expected.

### 4.2. Methodology and Performance Metric

Figure 2 shows the methodology in assessing the ML model. During data collection, the required data are gathered for the studied cells using the PM systems, as discussed previously. The data set analyzed in this research was collected from a Tier 1 Mobile Operator with quite a mature LTE network, which consists of more than 3000 base stations operating in various LTE frequency bands (800 MHz, 900 MHz, 1800 MHz, 2100 MHz, 2300 MHz and 2600 MHz).

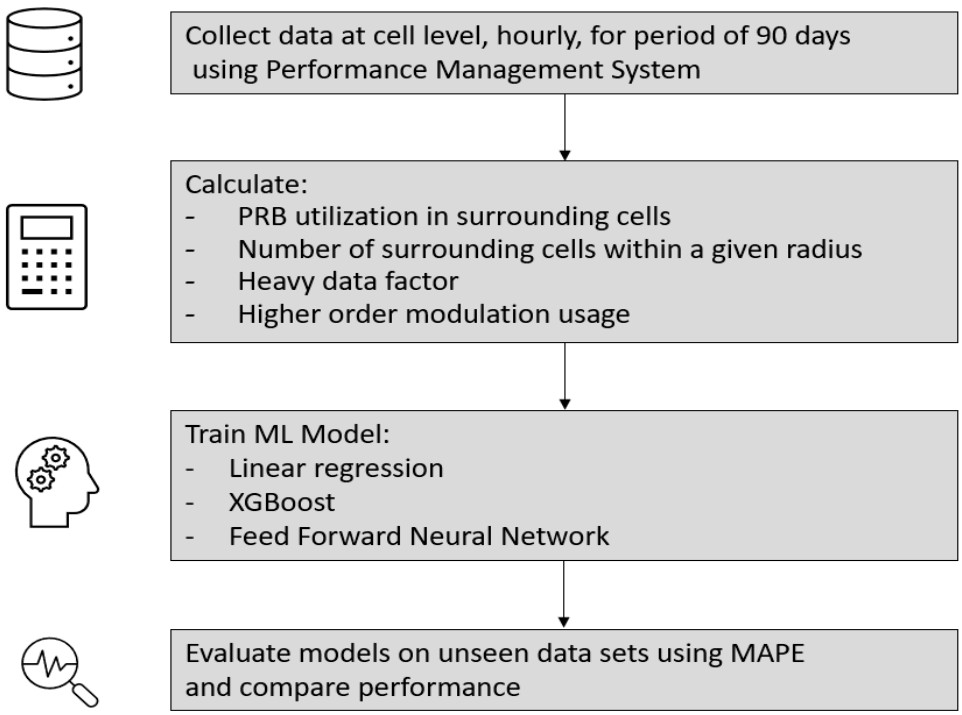

**Figure 2.** The workflow investigating alternative models.

Data were collected during a period of three months with hourly resolution, with more than 2000 records per cell. In pre-processing the PRB utilization in surrounding cells, the number of surrounding cells, the average PRB utilization in surrounding cells, the heavy data factor, and the higher order modulation are calculated from the raw data. The ML model being studied is trained to predict the average CQI using these inputs. During testing, the performance of the model is evaluated on unseen data. The performance is then compared to the other models.

The predictive performance of taking the mean CQI as the predicted value was assessed to establish a baseline naïve method. Next, the predictive performance of popular ML models including linear regression [18] (implemented using the sci-kit learn library [19]), extreme gradient boosting (implemented using XGBoost) [20,21], and feed-forward neural networks were compared. The neural network was implemented using the Keras library [22,23], feed-forward architecture, three hidden layers, 256 neurons per layer, Relu activation function, and Adam optimizer.

Mean Absolute Percentage Error (*MAPE*) was used as the performance metric throughout the study. *MAPE* was calculated according to the equation shown below.

$$MAPE = \frac{100}{N} \sum_{i=0}^{N} \left| \frac{predictedCQI_i - measuredCQI_i}{measuredCQI_i} \right| \tag{1}$$

Finally, the feature importance of the best performing is investigated. The features are scored by the number of times they are used to split nodes of trees in the ensemble. The features are then ranked by their score. Due to the stochastic nature of XGBoost, results may vary slightly.

## 5. Models Behavior and Feature Importance

The Pearson and Spearman Correlations of the four continuous features and average CQI are shown in Table 2. As can be seen in this table, surrounding PRB utilization has a slight but discernible correlation with average CQI. This correlation is stronger when taking total surrounding PRB utilization rather than the average. The Pearson and Spearman correlation coefficients are similar for all four features. This indicates the correlation is approximately linear.

**Table 2.** Correlation Coefficient of features.

| Feature | Pearson Correlation | Spearman Correlation |
|---|---|---|
| Average Surrounding PRB Utilization | −0.283 | −0.378 |
| Total Surrounding PRB Utilization | −0.369 | −0.427 |
| Higher Order Modulation | 0.74 | 0.752 |
| Heavy Data Factor | 0.404 | 0.409 |

Scatter plots of these four features against average CQI are shown in Appendix A (Figures A1–A4). A visual inspection of these plots also shows an approximately linear correlation. What can also be observed is that, in general, lower frequencies perform worse in terms of average CQI and achieved spectral efficiency, even for the same network load level captured as PRB Utilization with both considered metrics, due to radio propagation characteristics of different frequency bands and related inter-cell interference issues at lower frequencies, discussed earlier in Section 4.1.

A comparison of the mean absolute percentage error of the tested models is shown in Table 3. All three tested models are substantially better than the naïve method. The performance of the three models is similar. As the relationship between the average CQI and the indicators is simple and approximately linear, this is unsurprising. The best performing model is extreme gradient boosting. This model is used during the feature selection process.

**Table 3.** Model Performance.

| Model | MAPE |
|---|---|
| Naïve Model | 10.075% |
| Linear Regression | 5.857% |
| XGBoost | 5.507% |
| Feed-Forward Neural Net | 5.677% |

The time taken to train the models is shown in Table 4. Linear regression is the fastest, followed by XGBoost, and finally the neural network. Training time will vary slightly each run; however, the several order of magnitude differences between the models are unlikely to change. All experiments were run using an Intel i7-8565U CPU. It may be possible to train the neural network faster using a Graphics Processing Unit (GPU).

**Table 4.** Training Time.

| Model | Training Time (Seconds) |
|---|---|
| Linear Regression | 0.004 |
| XGBoost | 0.500 |
| Feed-Forward Neural Net | 100 |

During feature selection, XGBoost models trained using total surrounding PRB utilizations and average surrounding PRB utilization were compared. Average surrounding PRB utilization had a better performance, with a MAPE of 5.504% in comparison to 5.507%. This runs contrary to the correlation coefficient findings. It was found that eliminating any feature produced worse results. The MAPE using a definition of neighboring cells of 2.5 km, 5 km, and 10 km were 5.509%, 5.504%, and 5.577%, respectively.

The feature importance is shown in Figure 3 for the best performing XGBoost model.

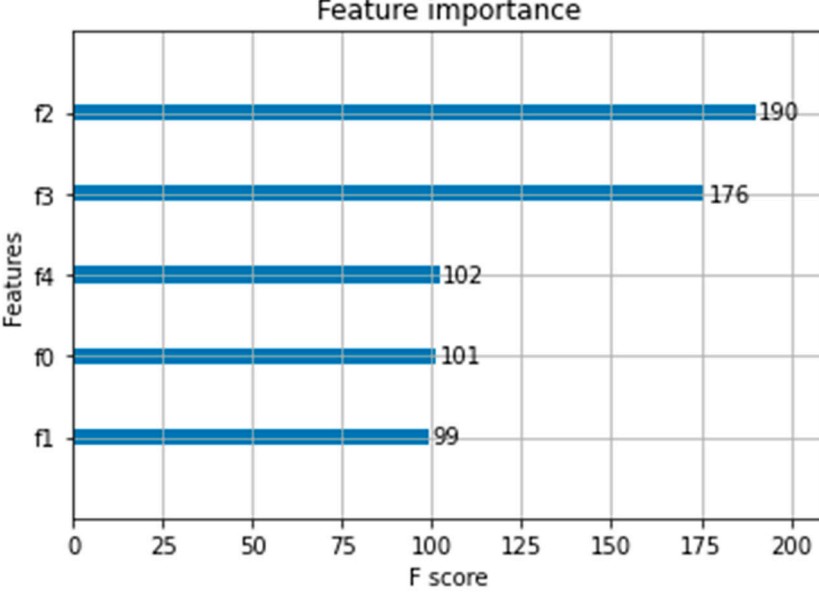

**Figure 3.** Feature importance.

The feature importance is found for a single decision tree by calculating the degree to which the split points using that feature improve the performance measure, when weighted by the number of samples being split. As XGBoost is an ensemble of decision trees it takes the average feature of importance across the trees [21]. The features in descending order of importance are the heavy data factor (f2), higher order modulation (f3), average surrounding PRB utilization (f4), frequency band (f0), number of neighbors (f1).

## 6. Discussion and Conclusions

The main contribution of conducted research is the application of ML techniques to problems of network design and capacity planning, and, in particular, spectral efficiency evolution modeling in growing network load conditions. Out of four tested ML models, XGBoost provided the best results and was selected for future work. The number of network performance indicators was introduced to the analysis, with the idea to improve modeling accuracy through the segmentation of network fragments.

A strong correlation is observed between CQI and PRB utilization, where a slightly stronger correlation was observed with PRB utilization when aggregated over surrounding cells; however, authors decided to introduce an average utilization and number of surrounding cells within the radius as separate features.

Higher order modulation usage shows the highest level of correlation with CQI, as expected due to the direct link between the two. It is introduced to capture information about user distribution with the reflection of general radio signal quality in a cell, and has a very high **F** score with a value of 176. The heavy data factor, intended to capture the type of the traffic mix in a cell in terms of burstiness, with a slightly lower level of correlation, shows the highest **F** score with a value of 190. The remaining three introduced features, frequency band, number of neighbors and average surrounding PRB utilization with an **F** score value of around 100, are also examined as very relevant features for model accuracy.

For the newly introduced feature number of surrounding cells within a radius, an optimal radius value of 5 km was estimated.

The continuation of the work and planned future activities are to test the model on a significant number of mobile networks, and investigate the impact of various network characteristics (topology, operating frequencies, maturity, design, traffic mix) on model behavior and accuracy.

## 7. Patents

System and method for determining user throughput in a cellular network, Aspire Technology, application number: PCT/EP2021/070902.

**Author Contributions:** Conceptualization, I.T. and P.I.; methodology, I.T. and E.B.; software, E.B.; validation, I.T. and E.B.; formal analysis, I.T. and E.B.; investigation, I.T.; resources, E.B.; data curation, I.T. and E.B.; writing—original draft preparation, I.T. and E.B.; writing—review and editing, I.T., E.B. and P.I.; visualization, E.B.; supervision, I.T. and P.I.; project administration, I.T.; funding acquisition, I.T. All authors have read and agreed to the published version of the manuscript.

**Funding:** This research received no external funding.

**Data Availability Statement:** The data presented in this study are available on request from the corresponding author. The data are not publicly available due to confidentiality issues and may be provided only with prior approval from data owner, which is the Mobile Operator that Aspire Technology supports on capacity planning activities. However, as PM systems and counters are quite unified, similar data can be fetched from any other mobile operator and analyzed with this methodology.

**Acknowledgments:** Authors would like to express gratitude to Aspire Technology Unlimited, for various type of support during the project.

**Conflicts of Interest:** The authors declare no conflict of interest.

## Appendix A

Figure A1 shows the correlation between Average CQI and higher order modulation. Each dot represents a single cell calculated over the duration of one hour during the day. Cells are segmented per operating frequency band; each band is represented as a different color. The impact of frequency can be clearly seen, where higher frequencies, for example, both carriers on 1800 MHz which are represented by red and green, tend to have a higher average CQI, while a lower frequencies, such as 800 MHz represented with blue, have lower average CQI values, which is in line with the theoretical expectations described in Section 4.

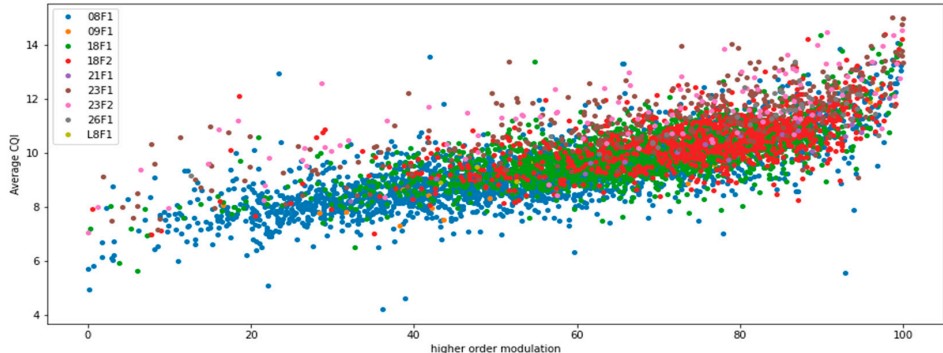

**Figure A1.** Higher Order modulation vs. Average CQI.

On the Figure A2, given at the top of the next page, the correlation between the Heavy Data Factor and Average CQI is analyzed. As discussed earlier, the main motivation for Heavy Data Factors' introduction was to segment cells with a traffic profile of more network demanding services, like video streaming or FTP download. For these services, the amount of data transmitted during TTI is higher. The only way to detect such behavior is to observe the average data payload per TTI. On the other hand, such an indicator is clearly driven, not just by offered traffic, but with achieved spectral efficiency on top. In line with expectations, cells with higher operating frequencies perform better in terms of average CQI, followed by the Heavy data factor.

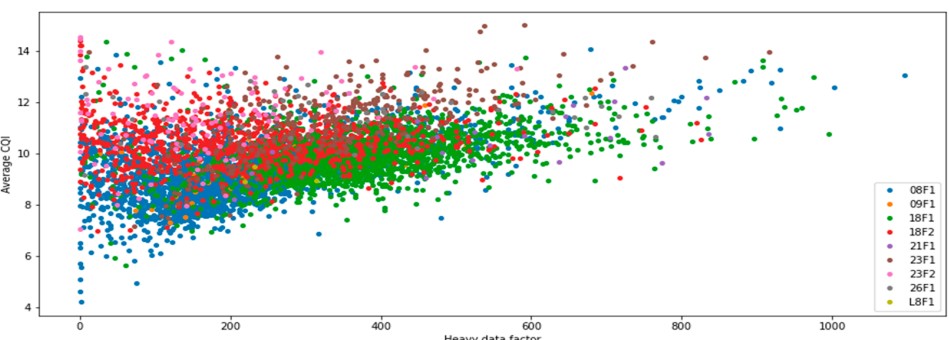

**Figure A2.** Heavy Data Factor vs. Average CQI.

The last two Figures A3 and A4, provide the correlation between Average CQI and PRB utilization in neighboring cells, captured by both metrics discussed earlier. In both cases, the trend of Average CQI decreasing as network utilization increases may be noticed. This is in line with expectations and may be explained by inter system interference increases. Lower frequencies perform worse with the same network load, due to radio propagation characteristics.

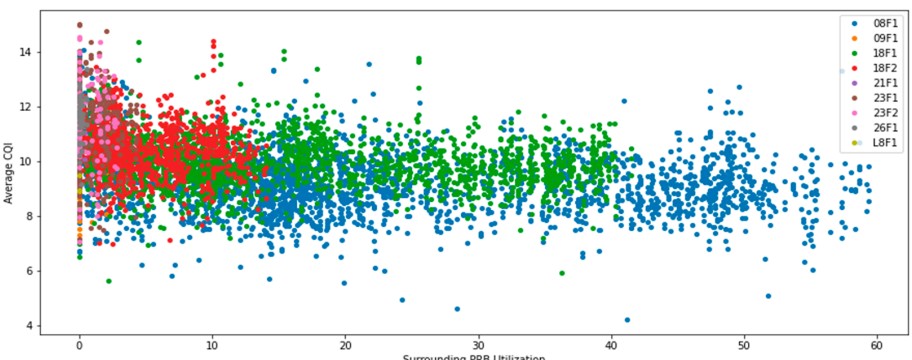

**Figure A3.** Total Surrounding PRB Utilization vs. Average CQI.

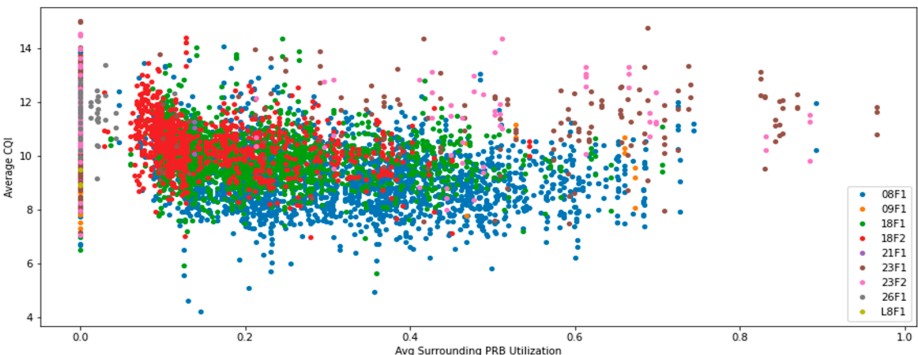

**Figure A4.** Average Surrounding PRB Utilization vs. Average CQI.

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
