# Peer review of "Predictive Capacity Planning for Mobile Networks—ML Supported Prediction of Network Performance and User Experience Evolution"

_electronics, doi:10.3390/electronics11040626_

Round 1

Reviewer 1 Report

  1. The methodology is confused - maybe some pseudocode or flowchart can be used to present methodology in more clear way.
  2. How exactly Mean Absolute Percentage Error was calculated?
  3. There is no information about the performance of used models - the cost of model training (complexity and/or time), performance in the sense of computation time.
  4. There is no information about evaluated ML models - implementation, input data, training and evaluation process.
  5. There is no detailed information about how computation experiments were performed.
  6. Results presented in Figures A1-A4 should be discussed in more detail.
  7. How about data used in computational experiments - how other researchers can obtain this data?

Author Response

We would like to thank the editors and reviewers for their comments and suggestions that have led to the improvement of the presentation of this paper. We have revised the manuscript to address all the comments of the reviewers.

Reviewer 2 Report

The manuscript is a significant topic.

  1. Analysis of previous studies on this topic needs to be added. It is necessary to check whether there are any studies using deep learning technology.
  2. The title of the chapter needs to be written in detail.
  3. For the readability of the manuscript, please explain the structure or flow diagram of the proposed model with pictures in Chapter 3.
  4. Please add an analysis of the most recent study most closely related to this study to Chapter 2 and References.

Author Response

(The authors gave the same response as above.)

Round 2

Reviewer 1 Report

I have no further comments.